# Zn-Pb Dumps, Environmental Pollution and Their Recultivation, Case of Ruda Śląska-Wirek, S Poland

**Miłosz Huber** [1,*] and **Olga Iakovleva** [2]

1 Department of Geology, Soil Science and Geoinformacy, Faculty of Earth Science and Spatial Management, Maria Curie–Skłodowska University, 2d/107 Kraśnickie Rd., 20-718 Lublin, Poland

2 Department of Applied Linguistics, Faculty of Humanity, Maria Curie–Skłodowska University, 5 Maria Curie–Skłodowska Sq., 20-033 Lublin, Poland

* Correspondence: milosz.huber@mail.umcs.pl

**Abstract:** This article describes the results of a study on the Zn-Pb heap, which is located in the center of the city of Ruda Śląska. The heap dates back to the 19th century but was rediscovered in the 21st century and abandoned. Located in the center of the city, it is eroded and contributes to the spreading of pollutants. The authors performed a study on the components of the dump using microscopic observations and geochemical analyses. The results indicate that the components of the heap are mobile, mainly due to the infiltration of meteoric waters affecting the contamination of soils and plants. The present work is devoted to a review of the state of the environment in the area of the heap and a proposal for its reclamation by covering it with an isolation layer or moving it to a protected place away from the city center. It is possible, in the future, to build an Environmental Education Center, for education and the monitoring of enrivonmental problems in Upper Silesia.

**Keywords:** Zn-Pb dumps; environmental study; recultivation; pollution; Ruda Śląska; S Poland

## 1. Introduction

In the Wirek district of Ruda Śląska, located in southern Poland [1–3], there are heaps that contain various types of waste from the processing of the Zn-Pb movement. In the Upper Silesia region, there was mining associated with Zn-Pb ores in the past [4,5]. These heaps were formed as a result of the accumulation of slags representing waste from the smelting process of the ores, which were located at this site in rocks of Triassic age and places of their weathering [4,6,7]. At the time of mining, these heaps were located beyond the residential perimeter, and after the ores were depleted, they were covered with turf and overgrown by vegetation characteristic of the surroundings. At the present time, as a result of the urbanization of the area, the heaps are located in the center of the city; moreover, due to the underground mining work carried out, the ground on which the heaps lie is subject to uneven settlement, forming fissures. One of them was uncovered in the early 2000s for re-excavation purposes and has since been abandoned, creating a water hazard and a nuisance for residents of the nearby buildings through the possibility of dust blowing from its surface (Figure 1). An additional problem is its poor protection from entry, as they are used by young people for extreme sports (motorcycle cross riding), and thus, further erosion of the heap. The purpose of this article is to inventory the condition of the heap, based on surveys conducted on site and on laboratory analysis, and to identify possible ways to protect it from further environmental degradation at the site.

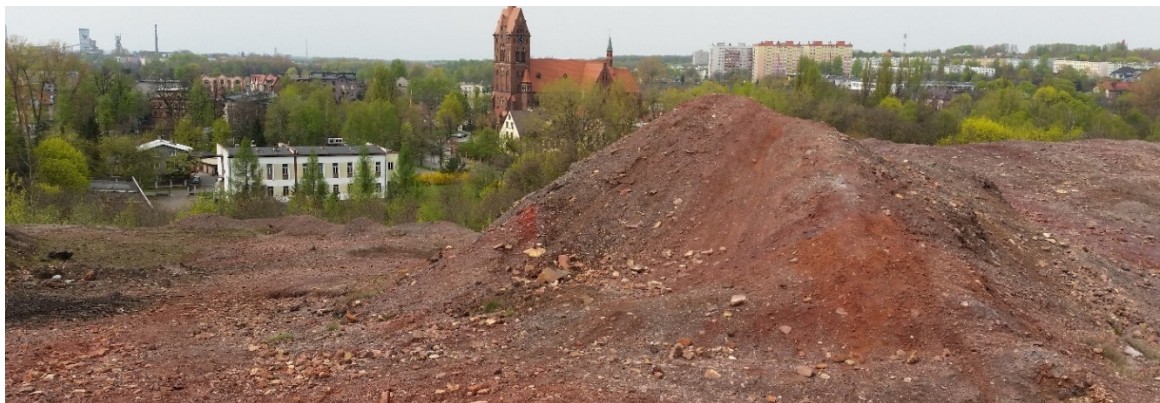

**Figure 1.** Landscape of an exposed heap in the center of the city of Ruda Śląska.

## 2. Information from Study Area

The heap in question is located in the area of Nawara Street and is situated in such a way that the street cuts the heap into two parts (Figure 2). The entire area under study is an approximate rectangle measuring 320 × 500 m. The northern part of the heap is currently covered with forest. The height of the trees there reaches 50 m. They are mainly common birch (*Betula pendula Roth*), common maple (*Acer platanoides* L.), ash maple (*Acer negundo* L.) and robinia (*Robinia pseudoacacia* L.). The southern part is only partially overgrown, adjacent to the street. Further away, behind a small ditch, there is an exposed heap (Figure 2). Samples for the study were collected in the area of the southern heap, where there are walls with an exposed, visible profile of the heap. Although the heap in question was mainly built in the 19th century [7], a detailed analysis of it indicates that younger waste including modern garbage is also present. At present, the heap is surrounded by buildings including residential houses; in some cases, local people in close proximity to the heap have cultivated gardens; and on the heap itself, wild resting places for the public are visible (traces of bonfires and bicycle and motocross "paths"). In some places, the heap is overgrown with sparse vegetation (self-sown trees), mainly common birch (*Betula pendula Roth*) [8].

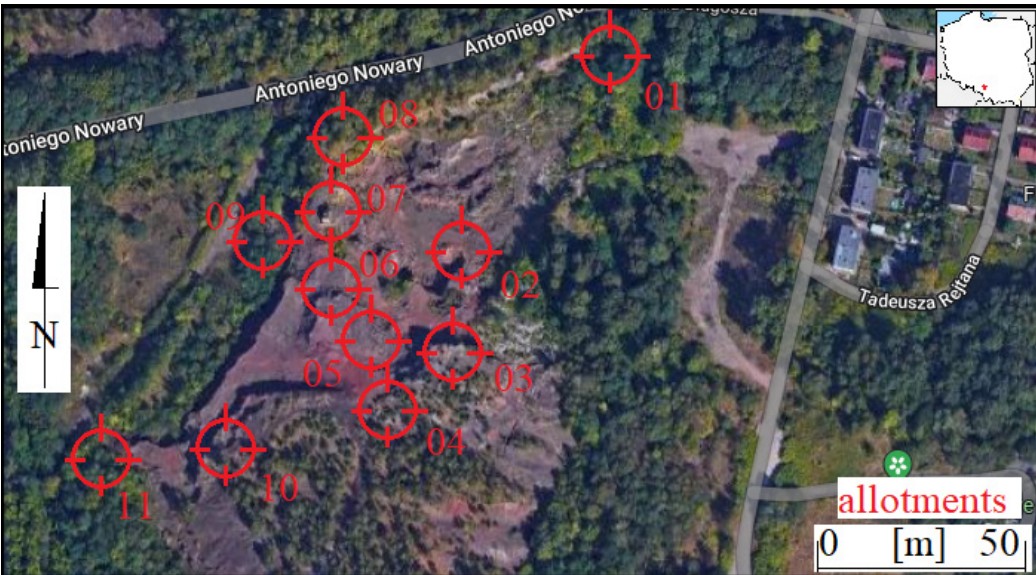

**Figure 2.** Location of sampling points for heap, soil and plants (on Google Maps).

The samples were taken in the southern heap at Nowara Street. The focus was on sampling open exposures in the heap that resulted from its stopped mining (points 02–04, Figure 3B,C). These sites contained fresh exposures of heap material. They also allowed

for the possibility of reviewing the vertical profile of the heap. Other sampling sites were located at the top of the heap (points 05, 06). At the top of the heap, perennial processes and plant succession were observed (Figure 3D). Samples were also collected in the periphery of the heap (points 01, 07–11, Figure 3A).

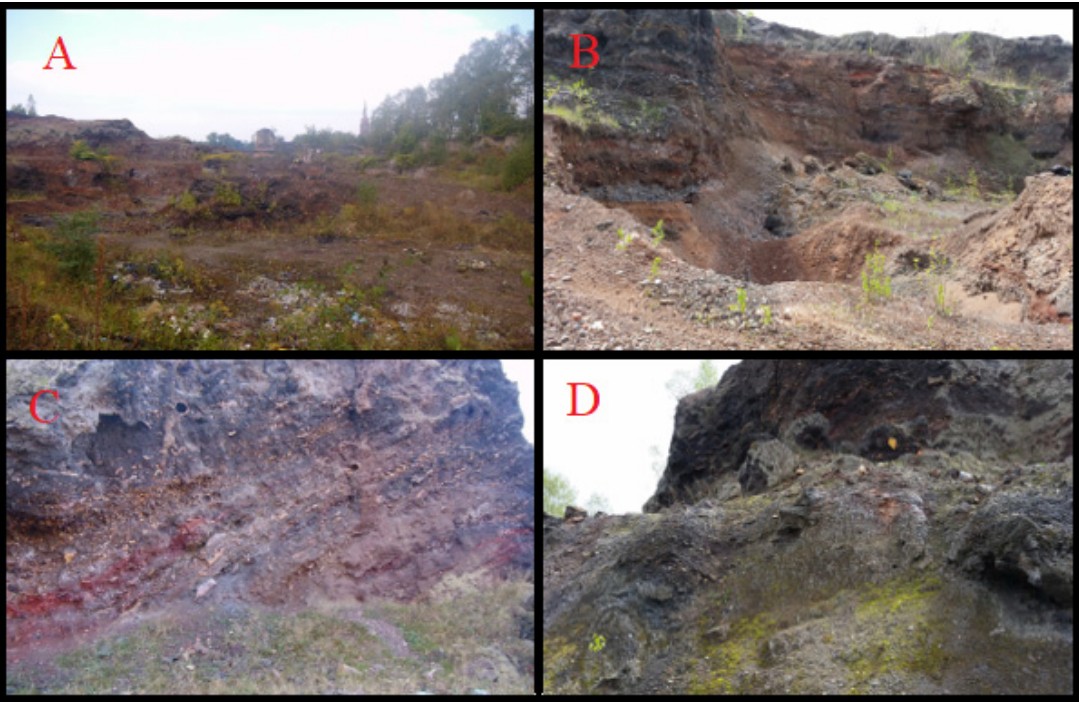

**Figure 3.** Photo documentation of the condition of the heap at the time of the survey: showing the exposure of the heap (**A**), its layered nature (**B**), fragments of ceramic metallurgical elements (**C**), mosses and lichens on the heap (**D**).

## 3. Materials and Methods

Samples of material from the heap were collected from several locations. Selected samples were used to make preparations of thin sections (0.001 mm in thickness) on polished, uncovered plates for further study. These slides were examined using a Leica DM2500P optical microscope in reflected and transmitted light, and then in the micro-area using a Hitachi SU6600 scanning electron microscope with an EDS attachment at the Department of Geology, Soil Science and Geoinformation at Maria Curie-Skłodowska University (UMCS) in Lublin. A total of 124 slag samples were examined, conducting analysis in 1709 micro-areas. Plants, which were preserved and dried after transport, were also sampled in the field. Observations of these plants were made macroscopically and under a binocular magnifying glass. They were also examined with an electron microscope to identify dust on their surfaces. A total of 424 analyses were performed on 12 plant samples. Geochemical investigations were accomplished using ICP-OAS (Varian). Selected samples were solubilized using a MARS microwave melter, then placed in an argon atmosphere and, afterwards, fed to the ICP, where elemental composition analysis took place. These studies occurred under the supervision of dr. Lesia Lata (UMCS). Sulfide studies were undertaken in $\delta^{34}$S from selected waste samples in the Institute of Physics in Maria Curie-Skłodowska University in Lublin. In turn, a $\delta^{34}$S stable isotopes analysis from whole separated sulfide minerals was performed at the Institute of Physics, UMCS, in Lublin. This was performed using a mass spectrometer with two inlets, triple collectors and $SO_2$ as the analyzed gas. The sulfide samples were first ground and dried and then weighed at an amount of about 20 mg, then mixed with sodium metaphosphate (60 mg) and cuprous oxide (80 mg) and rubbed together in a mortar. After heating the samples at a temperature of 200–300 degrees, they were cooled with liquid nitrogen and then reheated

to 850 °C, causing $SO_2$ evolution. Along with the analyzed samples, a 20 mg portion of the IAEA-S1 (Ag2S) standard was transformed into $SO_2$ and analyzed on a mass spectrometer and tested to normalize the obtained $\delta^{34}S$ values to the VCDT scale. These studies were performed by prof. Stanislaw Halas (UMCS). Visualization of the distribution of the studied pollutants was obtained using the Surfer program. The grid of points set according to GPS data was applied in the Surfer program to the metal content data, and then the program automatically plotted content maps and edge points, the zones beyond the study area were removed and the spatial extent was superimposed on the underlay (map).

## 4. Results

The conditions of the heaps in question vary. Located north of Nowara Street, the smaller heap is overgrown with vegetation, has developed mulch and shows no secondary changes (on the surface). The southern heap in question is in a much worse condition due to its stripped surface and partially exposed profiles and poses a major threat to the surrounding environment [9–24]. The subsections below cite the results of the analyses of slag, soil and plant samples taken from the southern heap.

### 4.1. Study of Heap Samples

Detailed studies of the heap profiles indicate that they contain slag formed as waste during the processing of Zn-Pb ores [8,25]. It now forms a compacted material and is mixed with the ceramic lining of the furnaces. Other waste associated with economic activities, such as coal and ash fragments, are also present. Macroscopically, the material is highly porous, spongy and brick-red in color. It has a varying degree of devitrified enamel with numerous bubbles in which air is generally present. Examination with a magnifying glass indicated that in some cases fragments of sulfides are embedded in the glaze, probably representing ore that was not completely melted in the smelting furnace. In the microscopic image, the heap material is a heterogeneous assemblage of mixed waste containing vitrified varieties of slag, residual ore minerals and new phases formed as a result of high-temperature processes in the blast furnace process, as well as secondary phases formed after the heap was deposited by water infiltration (Figure 4). Due to the highly porous nature of the slag and the heterogeneity of the heap, the sulfide minerals there represent different degrees of liberation. In the intact zones, this factor is relatively small, while it increases significantly in the zones of secondary fractures and cracks, where rainwater can freely migrate. There, precipitates of secondary phases (sulfates, carbonates and complex compounds) are formed. Due to the unstable nature of the subsoil (mining movements) and the partial mechanical exposure of the heap, there is a high risk that metal ions may be released throughout the area, increasing the coefficient of liberation.

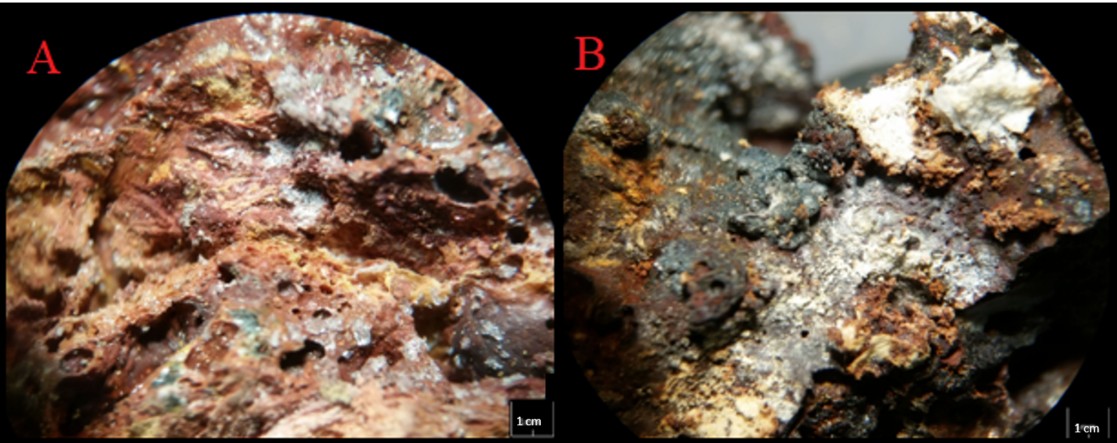

**Figure 4.** Examples of binocular magnifying glass microphotographs showing slag from the heap with visible enamel fragments (**A**) and precipitates (**B**).

In the slag material, pyroxenes and melilites can be identified, which were formed by annealing processes on the material. Glaze samples have a radial texture, resulting from devitrification processes and the presence of crystallites. They are highlighted by the presence of hematite and goethite, forming rims around the slag glaze. These minerals also surround the ore grains (galena, sphalerite, pyrite). The void zones contain secondary minerals classified as sulfates and oxides that form druse, radial structures, filling the voids in question (Figure 5). These minerals crystallized due to the migration of aqueous solutions, penetrating the material in question.

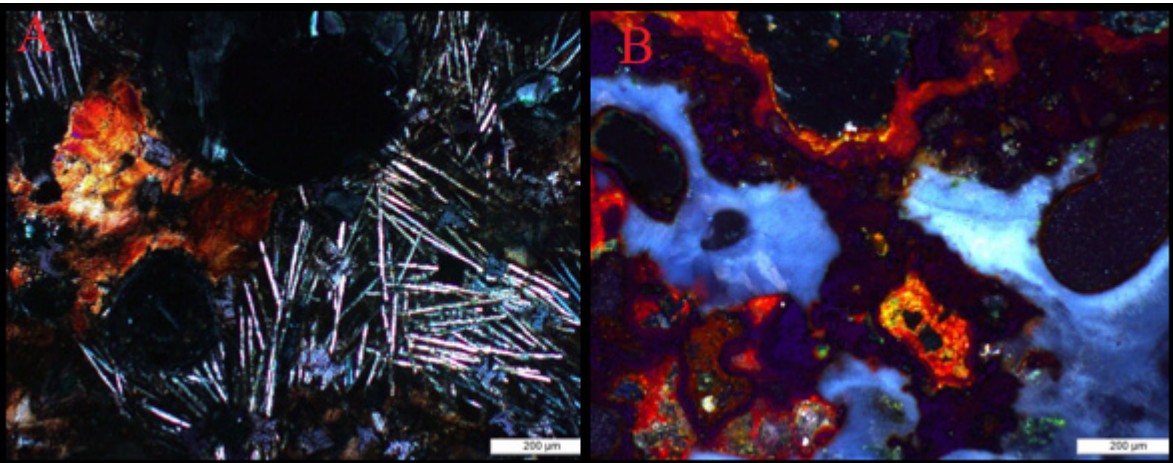

**Figure 5.** Microphotographs of slags with visible devitrified glaze (**A**) and iron oxides and hydroxides located in the background of the glaze and voids (**B**).

The micro-area studies were directed at tracing the phases involved in glaze devitrification, as well as identifying ore minerals and derived products associated with the migration of meteoric water in the slags in question. These studies have identified a certain diversity in these substances. Metals such as Zn and Pb with admixtures of Cu; Ti; and also Ni, As and Cd were found in these samples. Silicates were found in the glaze; they also formed their own phases, such as pyroxene and melilite. They were also accompanied by zinc silicates such as wilemite. In the case of ore minerals, mainly sulfides, such as pyrite, chalcopyrite, galena, sphalerite (or wurcite), were found, which were concentrated in the form of intrusions and epidotes within the slags (Figure 6). In addition to these minerals, metal alloys (mainly iron with admixtures of tin, zinc, copper) and polymetallic intrusions with lead, arsenic and cadmium in amounts sometimes reaching up to 30% by weight were found in the discussed minerals. Sulfates, carbonates and oxides were found in voids and efflorescence. Sulfates are mainly represented by gypsum (or barite) and carbonates by calcite. In addition to calcite, cerusite and smithsonite were also identified. Oxides of barium, zinc, lead and iron were also found located in the pores of the slags. In addition, as a result of the interaction of the solutions in the slag samples in question, complex compounds were formed, accompanying the phases in question. Along with the presence of hematite accompanied by iron hydroxides, becherite $((Zn,Cu)_6Zn2(OH)_{13}[(S,Si)(O,OH)_4]_2)$ was also found. These phases generally form epidote and efflorescence in the migration zones of water infiltrating the heap.

The slag samples were subjected to chemical analysis with ICP-MS. The results of these tests are listed in the table below (Table 1). The results indicate the presence of lead, zinc and copper but also significant amounts of arsenic, cadmium, and also chromium, titanium and nickel in these samples. With the volume of the heap in question being approximately 0.07 km$^3$ and the other, smaller with a volume of 0.03 km$^3$ (located north of Nowara Street), it can be assumed that the content of the metals in question in both heaps is approximately: for iron 44 Mt, manganese 3 Mt, copper and lead 1.8 Mt, chromium and nickel 0.1 t, and cadmium 0.01 t. These amounts are too small to be of interest for re-exploitation.

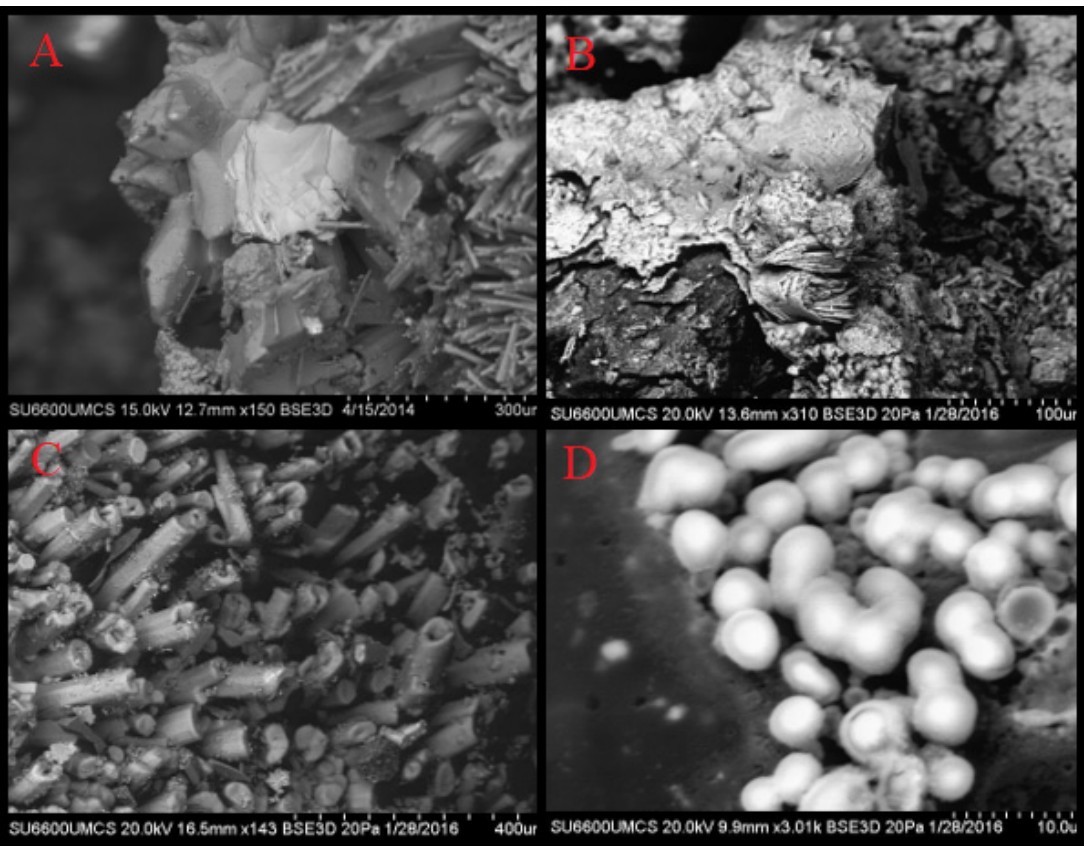

**Figure 6.** Backscattered electron (BSE) microphotographs showing the structure of the slag and the phases present: iron oxides and carbonate needles (**A**), copper, zinc and iron sulfides (**B**), gypsum crystals (**C**) and iron and zinc alloys (**D**).

**Table 1.** Results of chemical analysis of slag samples: content in [ppm] or in [wt. %] of determined *.

| Sample | Fe | Mn | Cu | Pb | Cr | Ni | Cd |
|--------|------|-------|--------|--------|-------|------|-------|
| 01Z | 47.7 * | 2.79 * | 0.19 * | 0.03 * | 47.34 | < | < |
| 02Z | 25.2 * | 184.7 | 0.02 * | 0.44 * | 25.74 | < | < |
| 03Z | 19.3 * | 0.28* | 0.14 * | 0.27 * | 40.81 | 60.9 | < |
| 04Z | 17.3 * | 4.30 * | 0.18 * | 0.20 * | 53.53 | 32.7 | < |
| 05Z | 8.85 * | 3.97 * | 0.17 * | 0.38 * | 52.39 | 53.6 | 19.6 |
| 06Z | 9.87 * | 831.4 | 0.10 * | 1.42 * | 22.91 | 33.3 | 51.4 |
| 07Z | 13.8 * | 269.5 | 0.03 * | 0.93 * | 53.83 | 16.1 | 59.4 |
| 08Z | 25.2 * | 651.2 | 0.15 * | 1.43 * | 57.51 | 80.3 | 49.7 |
| 09Z | 11.9 * | 1.32 * | 5.20 * | 0.11 * | 64.05 | 53.6 | 1.77 |
| 10Z | 7.20 * | 0.10 | 1.66 * | 2.53 * | 29.52 | 21.7 | 12.6 |
| Average contents: | 18.63 * | 1.29 * | 0.78 * | 0.74 * | 44.76 | 44.02 | 32.41 |

In order to determine the secondary processes taking place in the slags in question, the isotopic values of sulfur $\delta^{34}S$ were also measured: for samples 1–4, they were, respectively: 11.07‰, = 11.20‰, 5.37‰ and 1.21‰. The results of these analyses indicate two types of sulfur: neogene sulfur, which is the product of sulfophilic bacteria, and sulfide-derived sulfur. This means that we have both processes of a far-reaching biogenic nature and processes acting directly on the "ore" as a result of slag infiltration. Oxygen analysis showed an isotopic composition characteristic of rainwater infiltration.

### 4.2. Analysis of the Soils Samples

The surface of the heap in question was exposed due to mining operations. Soil has not yet covered its full volume. Usually, some accumulation of dusty fraction substances can be found on the open surface, occurring as a result of airborne (wind) transfer or as a result of snow nivation. In some places, patches of carried soil were preserved for protection. Initial soil formed by the crumbling of heap components is currently forming in many places, as confirmed by microscopic observations and the characteristic brick-red color of this soil. Analysis in the micro-area showed that they are analogous or homogeneous with the heap components. There are grains of quartz, feldspar, clay substances, carbonates and sulfates (including barite); admixtures of oxides and hydroxides of iron and manganese; and numerous metal elements such as zinc, lead and copper titanium (Figure 7, Table 2).

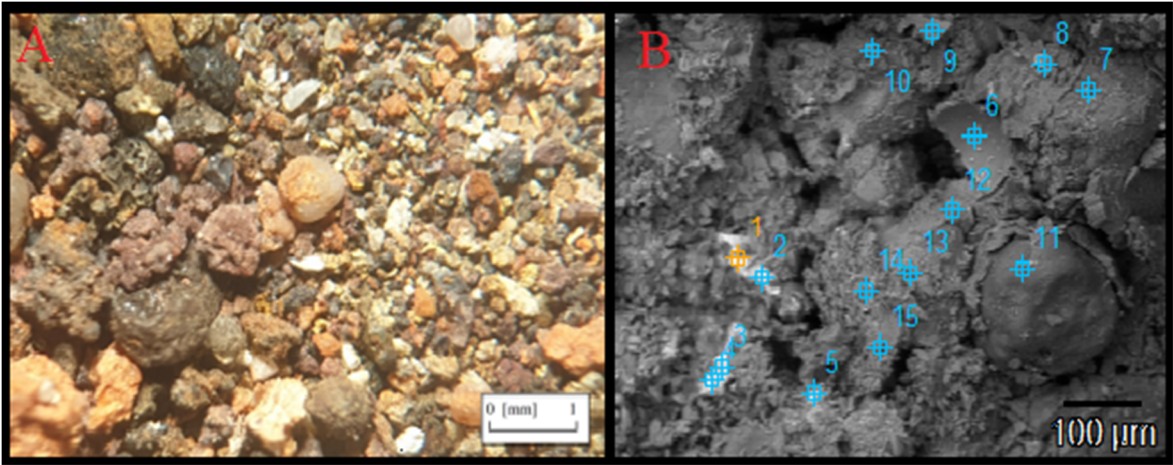

**Figure 7.** Microphotograph under reflected polarized light showing the grains of the initial soil taken from the heap (**A**) and microphotograph BSE of the soil sample with the points of microanalysis (**B**).

**Table 2.** Results of the chemical analysis of soils from the heap: content in [ppm] or in [wt. %] of the determined *.

| Sample | Fe | Mn | Cu | Zn | Pb | Cr | Ti | Ni | As | Cd | Hg |
|--------|------|--------|--------|-------|-------|-------|-------|------|-------|------|------|
| 1G | 4548 | 2.72 * | 61.85 | 69.12 | 31.39 | 40.55 | 312 | 3.98 | 13.25 | 0.74 | 0.04 |
| 2G | 2796 | 1.72 * | 40.82 | 18.06 | 3.82 | 20.96 | 252.8 | 4.15 | 10.11 | 0.59 | 0.04 |
| 3G | 3427 | 1.96 * | 57.05 | 73.16 | 38.73 | 28.92 | 274.3 | 3.19 | 11.22 | 0.75 | 0.03 |
| 4G | 8207 | 2.96 * | 176.90 | 57.32 | 31.62 | 45.73 | 264.9 | 7.05 | 13.56 | 0.81 | 0.23 |
| 5G | 4105 | 2.94 * | 57.56 | 26.73 | 7.60 | 37.88 | 262.1 | 3.39 | 13.93 | 0.57 | 0.02 |

Chemical tests of soil samples taken from the heap indicate that concentrations of metals such as copper, nickel, lead and zinc are still high. Critically high levels of arsenic and cadmium were also found in these samples (Table 2).

### 4.3. Results of the Plants Sample Analysis

Field studies have found that, due to the porous nature of the heap components, many plants grow directly on this substrate (this is especially true of mosses and lichens).

Shrubs and tree seedlings usually utilize crushed material and accumulations of dusty substances; their root system also takes advantage of cracks in the heap, formed due to uneven ground settlement, and its destabilization as a result of mining and exposure of batches of heap material. In the course of the field survey, samples of firefly moss (*Polytrichum commune Hedw.*) were taken from 10 sites, as well as lichen (*Cladonia* sp.) and the leaves of the common birch (*Betula pendula Roth*) at one site (Figures 2 and 8).

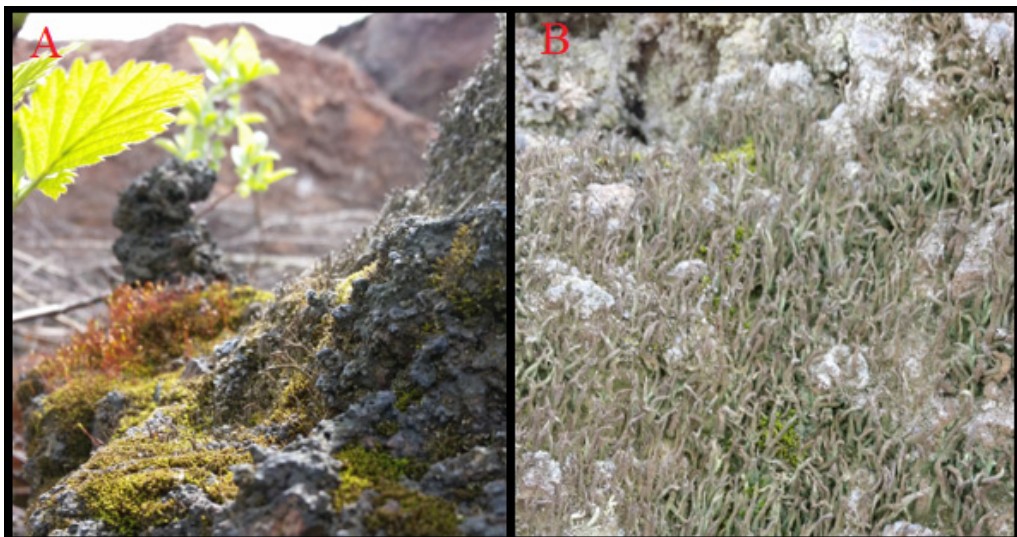

**Figure 8.** An example of firefly moss (*Polytrichum commune Hedw.*, **A**) and lichen (*Cladonia sp.*, **B**) growing on a heap.

The micro-area studies were conducted to identify the dust that was on the surface of the herbaceous parts of the plants in question. These tests revealed admixtures of metals such as iron and, in smaller amounts, titanium, zinc, lead, arsenic, copper, vanadium and barium. The concentration of metals in the studied samples fluctuated mainly at the level of parts of a percent by weight, rarely exceeding 0.5%, and in extreme cases, reaching values above 4–5%. The performed ICP-OES analyses in the heap area of nine moss samples (labeled H1–11) and one lichen sample (labeled H3p) and also one comparison sample, taken from the center of Ruda Śląska at Katowicka Street (in the park, labeled RS03), unequivocally showed that the metal content was elevated in all samples. Zinc values ranged from a few hundred ppm to over a thousand, similarly for lead, whose content in the tested plants was sometimes 304 times higher (Table 3). An admixture of copper and nickel was found in the range of 20–426 ppm. A high content of cadmium and arsenic, sometimes reaching several hundred ppm, was also found.

**Table 3.** Summary of the results of ICP-OES analysis of moss samples and one sample taken from the lichen (3PL*) on the heap in Ruda Śląska-Wirek [ppm].

| Sample | Zn | Mn | Ni | Cu | Cd | Fe | Pb | As | Ti |
|--------|------|--------|------|-------|-------|--------|---------|--------|------|
| 1P | 404 | 65.5 | 25.8 | 20.2 | 1.55 | 1248 | 66.23 | 6.95 | 159 |
| 3P | 955 | 146.0 | 22.4 | 38.4 | 3.95 | 4104 | 384.00 | 71.90 | 235 |
| 3PL* | 862 | 184.0 | 33.5 | 93.1 | 15.60 | 25,000 | 1258.00 | 893.00 | 981 |
| 5P | 923 | 492.0 | 60.8 | 426.0 | 17.80 | 42,267 | 3645.00 | 508.00 | 1863 |
| 6P | 832 | 373.0 | 35.9 | 81.5 | 5.42 | 11,617 | 111.00 | 209.00 | 804 |
| 7P | 771 | 256.0 | 24.5 | 26.1 | 3.03 | 10,639 | 317.10 | 88.50 | 625 |
| 8P | 1486 | 1054.0 | 97.9 | 124 | 8.11 | 28,551 | 1199.00 | 172.00 | 1588 |
| 9P | 231 | 69.4 | 26.3 | 35.4 | 1.44 | 1670 | 79.80 | 14.20 | 273 |
| 10P | 1440 | 784.0 | 49.5 | 80.7 | 8.91 | 14,846 | 481.80 | 117.00 | 918 |
| 11P | 1082 | 63.6 | 32.4 | 59.6 | 10.10 | 3184 | 896.10 | 34.90 | 249 |
| 03RS | 810 | 809.0 | 38.6 | 42.2 | 4.29 | 10,573 | 536.40 | 31.50 | 693 |

The proportions of these metals fluctuated in different plants but were generally significantly overstated. The content of these metals is also high in the comparison sample from the center of the city, indicating high pollution in the entire area.

## 5. Discussion

The examined heap material has numerous admixtures in the form of polymetallic inclusions, fragments of ore minerals and various precipitates, found by microscopic examination. Sulfides in the formations in question pose a particular threat, as they can easily oxidize and become mobile. They also contribute to the acidification of water migrating in the heap and the formation of precipitates [1,2,25–30]. Occurring admixtures of cadmium and copper in sulfides are encountered occasionally, but due to the toxicity of the elements, they pose a major threat to the environment. The precipitation dissolving these compounds can contribute to their migration and accumulation in water, plants and, thus, in animals living in the area [31]. Some of these elements are in oxidized form and as salts (sulfates and carbonates). Elements such as arsenic that form water-soluble hydroxides can be the most dangerous. In turn, lead, zinc and chromium form water-soluble sulfates. Studies of the slag have shown that it is highly porous and spongy, which allows rainwater to migrate through it. It also facilitates uneven ground settlement as a result of current mining operations at greater depths (the slag heaps are located in the overburden of a coal mine field) [32–38]. This causes the formation of zones enriched in precipitates and characterized by white efflorescence on fragments of exposed slag. The environmental nuisance of these metals is known and widely described by many researchers [39–48].

The distribution of heavy metal elements on the surface of the slag heap is not uniform. In the case of As, Cu, Cd and Pb, it can be concluded that the highest concentrations of these elements are located in the central part of the heap (Figure 9). The distribution of Ni shows higher concentrations in the vicinity of Nowara Street and, to a lesser extent, in the south of the heap. The same is true for zinc [38].

At the moment, the most important step would be to protect the heap from further erosion and dusting by applying soil to its surface. The processes of soil subsidence and cracking of the heap can be controlled on an ongoing basis, and fills can be used with impermeable material (clay rocks, clays) to prevent water from entering the heap. It may be worth considering moving the heaps to specially designated and protected sites where they can be neutralized [33], such as old mine galleries where Zn-Pb ores were mined where these elements occur naturally in the surrounding rocks.

The above-submitted survey results indicate that the heap site is a hazardous post-industrial site that is currently unprotected and exposed (dug up). The impact of forming dust slurries from the heap is legible in the area. Wind can blow the heap material containing heavy metals long distances. The poor security of the heap is a serious problem. Basically, today, anyone can enter; young people organize cross trails and there are numerous places indicating the organization of recreation on the heap (traces of bonfires, barbecues, Figure 8). Another problem is the nearby allotment gardens, located in close proximity to the heap or on the part of the heap covered with topsoil (Figure 8). Even if the inside of the heap is not visible locally, edible plants can become enriched in the heavy metals found there through the root system.

An interesting proposition is the possibility of building an Environmental Education Center near the heap with properly prepared laboratories. Building such a facility would allow monitoring the condition of the heap in real time. The study of the heap and its settlement and infiltration processes could also result in interesting scientific discoveries in the future, which could be applied to other heap sites in Upper Silesia. An added value of the site will be the possibility of conducting numerous educational programs, making residents aware of the value of the environment and ways of fighting pollution and garbage [49–54]. Conducting educational programs and the exhibition–museum could also economically revitalize the region of Ruda Śląska near the heap. It would also have a positive impact on the image of the district, developing an area currently excluded from traffic. The activity of the center could also influence the management of the residents' time, especially those who currently spend their free time on the heap, which poses a threat to their health and contributes to further erosion of the heap. The discussed solution may be cost-intensive, but there is currently a lack of such facilities in the Upper Silesia region

and, given the mining nature of the region and the numerous operations that are also being carried out today, it could be a justifiable measure.

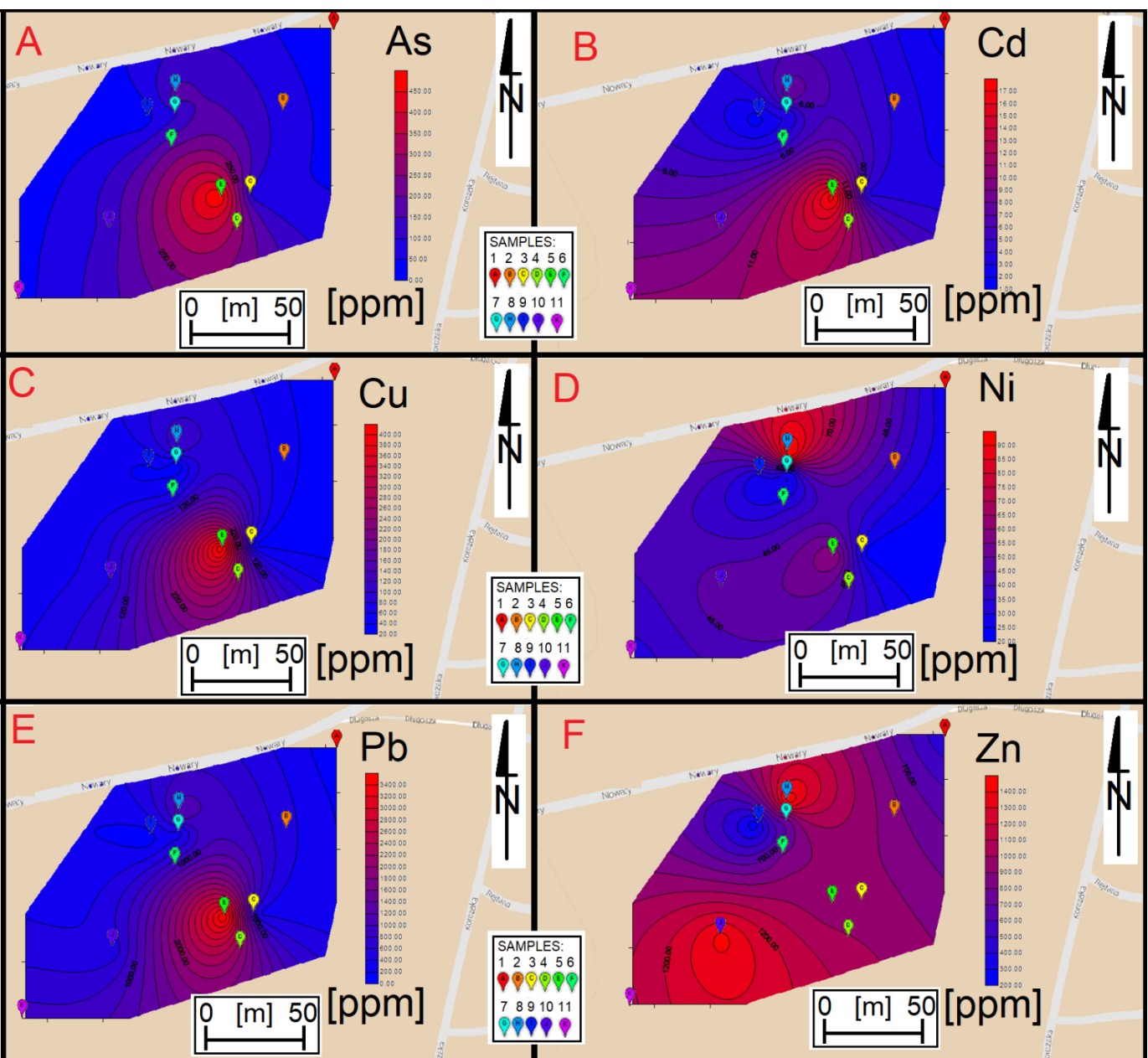

**Figure 9.** Maps showing the distribution of heavy metal elements in the heap area Maps were made respectively for: arsenic (**A**), cadmium (**B**), copper (**C**), nickel (**D**), lead (**E**) and zinc (**F**).

## 6. Conclusions

The results presented above indicate that the heap in Ruda Śląska currently poses a major threat to the environment. The pits are exposed, unprotected, washed out by rain and blown by the wind. This is where the highest concentrations of heavy metal (Cd, Cu, Pb) contamination occur. In this form, the heap is dangerous to the environment. Another problem is the uneven settlement of the ground under the heap, creating gaps into which water migrates, flowing through the heap and washing out the metals it contains. The heap should be protected from further erosion by covering it with a layer of impermeable soil. Attention should also be paid to the awareness of local residents to avoid creating allotment gardens in the immediate vicinity of the heap and spending time directly on the heap. The

entrance to the heap should be protected by a fence with information boards placed to prevent the free movement of outsiders on the heap and its erosion due to extreme sports on the heap. These activities may cause the heap to become unsealed again. The vicinity of the heap, once secured, should be monitored on an ongoing basis, and if leaks are found in the secured heap, it may be necessary to relocate it to another secured location. In the future, an Environmental Education Center could be built in the area. The center would have an educational role, building environmental awareness among residents and at the same time would perform monitoring of the heap. In a broader context, such a center could also deal with the issue of reclamation of other similar heaps, of which there are a significant number in the Upper Silesia region.

**Author Contributions:** Conceptualization, M.H. and O.I.; methodology, M.H.; software, M.H.; validation, M.H. and O.I.; formal analysis, M.H.; investigation, M.H.; resources, M.H.; data curation, M.H. and O.I.; writing—original draft preparation, M.H.; writing—review and editing, M.H. and O.I.; visualization, M.H.; supervision, M.H.; project administration, M.H. All authors have read and agreed to the published version of the manuscript.

**Funding:** This research received no external funding.

**Data Availability Statement:** Not applicable.

**Acknowledgments:** The authors would like to thank Lesia Lata for performing geochemical studies of soils and plants and Stanisław Halas for isotopic analyses.

**Conflicts of Interest:** The authors declare no conflict of interest.

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
