# Peer review of "Zn-Pb Dumps, Environmental Pollution and Their Recultivation, Case of Ruda Śląska-Wirek, S Poland"

_mining, doi:10.3390/mining2030033_

Round 1

Reviewer 1 Report

This article analyzes a heap in the middle of a town. It describes the use, vegetation and, finally, composition of this heap.

It is understandable and it has enough references.

I suggest some improvements in the article:

- Methods: It is needed a description about Figura 9 is done.

- All the figures need more clearly and scale and north direction.

- Results: firts paragraph must be revised, because, it says some things that are already explained.

- Discussion: first paragraph, there is lot of information and suggestions but there is not any article that support this.

Author Response

Response to the first reviewer

Dear reviewer, thank you for evaluating our text, your time and comments. We respond below in the same convention:

- Methods: It is needed a description about Figura 9 is done.

Information on how figure 9 was made is given in the methodology (detailed).

- All the figures need more clearly and scale and north direction.

In all the figures, scales and land orientations were added.

- Results: first paragraph must be revised, because, it says some things that are already explained.

Thank you for your valuable attention, this paragraph has been revised.

- Discussion: first paragraph, there is lot of information and suggestions but there is not any article that support this.

Dodano niezbędną literaturę do niniejszego akapitu,

Autorzy jeszcze raz dziękują recenzentowi za wnikliwą analizę tekstu. Mamy nadzieję, że nasze odpowiedzi i załączony manuskrypt zadowolą recenzenta.

Reviewer 2 Report

This paper studies the negative influence on solid and water source caused by Zn-Pb dumps in the center of Ruda Slaska. My detailed review is as follows.

1.       The size of the thin, polished, uncovered plates is not given in the artical. Please add.

2.       As the important component of samples, more microphotographs about silicates could be given.

3.       The result of oxygen anylasis is not given in 4.1 study of heap samples.

4.       The recent publications are missing: In this study, the citrate-coated nanosized Ag paste was utilized to generate robust bare Cu–Cu joints under atmospheric conditions. 10.1016/j.pnsc.2020.12.004 

Author Response

Response to the second reviewer:

Dear Reviewer, Thank you for reading this article and for your valuable comments.

  1. the size of the thin, polished, uncovered plates is not given in the artical. Please add.

This information has been added.

  1. As the important component of samples, more microphotographs about silicates could be given.

The authors did not want to focus on silicates, since in their opinion these compounds are relatively well resistant to weathering and aqueous conditions. Therefore, the focus was rather on the documentation of sulfides, sulfates, carbonates and complex compounds.

3 The result of oxygen anylasis is not given in 4.1 study of heap samples.

Dear reviewer, with great regret we have to say that at the present time we no longer have the ability to accurately determine this result. This research was performed by Prof. Stanislaw Halas, who is currently deceased. The authors checked the obtained results in their archives but did not find these data, except for the professor's interpretation. We are very sorry, but we are unable to detail this research at the moment. Perhaps in the future they will be repeated.

4 The recent publications are missing: In this study, the citrate-coated nanosized Ag paste was utilized to generate robust bare Cu-Cu joints under atmospheric conditions. 10.1016/j.pnsc.2020.12.004

This has already been revised and references have been renewed....

The authors once again thank the reviewer for his insightful analysis of the text. We hope that our answers and the attached manuscript will satisfy the reviewer. We are aware that not all responses may have been fully satisfactory, but we kindly ask for your understanding.

Reviewer 3 Report

I read the manuscript thoroughly. A common method was used to identify components of a tailing dump, but there are some basic works to improve the article. However, the below are some comments and questions that I believe will help to enhance the manuscript.

-          Introduction part need a comprehensive revision in aspect of similar works’ methodologies and results.

-          Sampling pattern should be explained in detail.

-          The reserve estimation of dump is recommended to carry out in order to evaluate the processing probability of valuable minerals.

-          Due to the presence of sulphide minerals, especially pyrite, the phenomenon of AMD producing seems certain. This issue is necessary to be discussed by researchers.

-          Page 3, Fig 2, was not cited appropriately in text.  

-          More evidence is needed to identify the minerals indicated in Fig.6.

-          Fig. 7A, does not give clear information. The lack of clarity should be corrected.

-          Generally, there are a good relationship between soil contents and plants types, so, the authors could discuss about this issue.

-          Page 9, line 257, the number of references cited is not reasonable and some more generated inconvenience from hazardous components is necessary to be mentioned here.

-          The environment (software) used to obtain results in Fig.9 is not obvious.

-          A clear and concise conclusion is required to describe applied methods and findings clearly. The resources of contaminations as well as their distribution pattern should be clarified.

Author Response

Response to the third reviewer.

Dear reviewer, thank you for reading the non-essential text valuable comments to which we respond in the points below.

-          Introduction part need a comprehensive revision in aspect of similar works’ methodologies and results.

Authors leaned into text, added new work results in text.

-          Sampling pattern should be explained in detail.

This is a very valuable point and this text has been introduced in the methodology.

-          The reserve estimation of dump is recommended to carry out in order to evaluate the processing probability of valuable minerals.

Such calculations have been introduced.

-          Due to the presence of sulphide minerals, especially pyrite, the phenomenon of AMD producing seems certain. This issue is necessary to be discussed by researchers.

In the present heap, sulfates are a more significant problem, as these often form precipitates and are triggered when the geochemistry of water migrating through the rocks changes. Sulfides including pyrite (although not only pyrite is present there) are generally confined by silicates, which hinders their mobility.

-          Page 3, Fig 2, was not cited appropriately in text.  

Figure 2 was better quoted in the text.

-          More evidence is needed to identify the minerals indicated in Fig.6.

The authors have the results of optical and electron microscope observations. In addition, our research was not so much directed at phase identification (such analyses were still made by many other researchers earlier) but at the current state of the heap and the resulting risks. Hence, the authors tried to focus on this issue, citing literature where there are data on the mineralogy of these heaps.

-          Fig. 7A, does not give clear information. The lack of clarity should be corrected.

Fig. 7a has been corrected. We apologize for the blurry photograph.

-          Generally, there are a good relationship between soil contents and plants types, so, the authors could discuss about this issue.

Yes, this issue is very interesting and we intend to address it further in the future.

-          Page 9, line 257, the number of references cited is not reasonable and some more generated inconvenience from hazardous components is necessary to be mentioned here.

The citations have been completed. Thank you very much for bringing this to our attention.

-          The environment (software) used to obtain results in Fig.9 is not obvious.

More has been added about the software in the methodology.

-          A clear and concise conclusion is required to describe applied methods and findings clearly. The resources of contaminations as well as their distribution pattern should be clarified.

The ending has been rewritten.

The authors once again thank the reviewer for his careful analysis of the text. We hope that our responses and the attached manuscript will satisfy the reviewer. We are aware that not all responses may have been fully satisfactory, but we kindly ask for your understanding.

Round 2

Reviewer 3 Report

I studied the authors' responses. I think that after minor revisions (below comments), the corrected form can be acceptable to publish.

In connection with the hindering role of silicates in AMD generation, more evidence, e.g. liberation degree, are needed to provide.

Phase identification is very crucial in such studies. So, I recommend to add some more data to clarify the involved components.

Author Response

Dear Reviewer, After reading the comments, I decided to add the following paragraph to the text:

Due to the highly porous nature of the slag and the heterogeneity of the heap, the sulfide minerals there represent different degrees of liberation. In the intact zones, this factor is relatively small while it increases significantly in the zones of secondary fractures and cracks, where rainwater can freely migrate. There, precipitates of secondary phases (sulfates, carbonates and complex compounds) are formed. Due to the unstable nature of the subsoil (mining movements) and the partial mechanical exposure of the heap, there is a high risk that metal ions may be released throughout the area, increasing the coefficient of liberation.”

I hope this will be satisfactory. I can't give a strict value because I don't have sufficiently precise data at the moment so I determined it indicatively based on observations in the micro-area of the heap samples.